# On Generalization Bounds for Neural Networks with Low Rank Layers

**Andrea Pinto**                                                        PINTOA@MIT.EDU
*Center for Brains, Mind, and Machines, Massachusetts Institute of Technology*

**Akshay Rangamani**                                         AKSHAY.RANGAMANI@NJIT.EDU
*Department of Data Science, New Jersey Institute of Technology*

**Tomaso Poggio**                                                        TP@CSAIL.MIT.EDU
*Center for Brains, Mind, and Machines, Massachusetts Institute of Technology*

**Editors:** Gautam Kamath and Po-Ling Loh

## Abstract

While previous optimization results have suggested that deep neural networks tend to favour low-rank weight matrices, the implications of this inductive bias on generalization bounds remain underexplored. In this paper, we apply a chain rule for Gaussian complexity (Maurer, 2016a) to analyze how low-rank layers in deep networks can prevent the accumulation of rank and dimensionality factors that typically multiply across layers. This approach yields generalization bounds for rank and spectral norm constrained networks. We compare our results to prior generalization bounds for deep networks, highlighting how deep networks with low-rank layers can achieve better generalization than those with full-rank layers. Additionally, we discuss how this framework provides new perspectives on the generalization capabilities of deep networks exhibiting neural collapse.

**Keywords:** Gaussian complexity, Generalization bounds, Neural collapse, Low rank layers

## 1. Introduction

Deep learning has achieved remarkable success across a wide range of applications, including computer vision (He et al., 2016; Simonyan and Zisserman, 2014), natural language processing (Vaswani et al., 2017; Brown et al., 2020), decision-making in novel environments (Silver et al., 2016), and code generation (Chen et al., 2021), among others. Understanding the reasons behind the effectiveness of deep learning is a multi-faceted challenge that involves questions about architectural choices, optimizer selection, and the types of inductive biases that can guarantee generalization.

A long-standing question in this field is how deep learning finds solutions that generalize well. While good generalization performance by overparameterized models is not unique to deep learning—it can be explained by the implicit bias of learning algorithms towards low-norm solutions in linear models and kernel machines (Bartlett et al., 2020; Muthukumar et al., 2020)—the case of deep learning presents additional challenges. However in the case of deep learning, identifying the right implicit bias and obtaining generalization bounds that depend on this bias are still open questions.

In recent years, Rademacher bounds have been developed to explain the complexity control induced by an important bias in deep network training: the minimization of weight matrix norms. This minimization occurs due to explicit or implicit regularization (Neyshabur et al., 2015; Bartlett et al., 2017; Poggio et al., 2020; Xu et al., 2023). For rather general network architectures, Golowich et al. (2018) showed that the Rademacher complexity is linear in the product of the Frobenius norms

of the various layers. Although the associated bounds are usually orders of magnitude larger than the generalization gap for dense networks, very recent results by Galanti et al. (2023) demonstrate that for networks with structural sparsity in their weight matrices, such as convolutional networks, norm-based Rademacher bounds approach non-vacuity.

An alternative hypothesis suggests that the correct implicit bias for stochastic gradient descent (SGD) may be towards low-rank weight matrices. It is empirically well established that training of deep networks induces a bias towards low rank of the weight matrices (Galanti et al., 2022; Huh et al., 2023). In addition to such a bias for linear networks (in the matrix factorization problem (Gunasekar et al., 2017; Ji and Telgarsky, 2020)), an empirical study by Gur-Ari et al. (2018) found that during minimization SGD spans a small subspace, implying an effective bias on the rank of the weight matrices. Additionally, Timor et al. (2022) discovered that in sufficiently deep ReLU networks, when fitting the data, the weight matrices in the topmost layers become low-rank at the global minimum. This observation is also related to the property of Neural Collapse (Papyan et al., 2020; Xu et al., 2023; Rangamani et al., 2023) where the features and weights in the top layers of a deep network "collapse" to low rank (rank 1 in the case of binary classification) structures. However, we are not aware of a generally accepted theoretical statement explaining the role of low rank in generalization.

**Our contributions.** We compute the Gaussian complexity (Bartlett and Mendelson, 2002) of deep networks with rank-constrained layers and obtain a bound on the generalization error that takes advantage of rank-based complexity control. Our key insight is that by applying Maurer's chain rule for Gaussian complexity Maurer (2016a) (recently extended to Rademacher complexity (Chu and Raginsky, 2023)), we can avoid rank and dimensionality factors multiplying across layers in deep networks. As a result, we obtain a generalization bound for rank-constrained deep networks that depends on the rank $r$, depth $L$, and width $h$ as $\mathcal{O}\left(\prod_{i=1}^{L} \|\mathbf{W}_i\|_2 C_1^L L r \sqrt{\frac{h}{m}}\right)$. For the class of rank-constrained deep networks, our bound improves upon the size-independent generalization bounds obtained by Golowich et al. (2018).

**Outline.** We first introduce a few basic tools of statistical learning theory in section 2, and define the class of spectral norm and rank constrained deep networks that we are primarily interested in. In section 3 we show how existing proof techniques when applied to the class of rank constrained functions, do not yield bounds that depend on the rank. Our main results are presented in section 4. We discuss how our results compare to other norm based generalization bounds in section 5 including their strengths and limitations. We conclude in section 6.

## 2. Preliminaries

**Base settings.** We consider the problem of training deep classifiers mapping inputs in $\mathcal{X} \subseteq \mathbb{R}^d$ to labels in $\mathcal{Y} \subseteq \mathbb{R}^C$. We are studying deep neural networks which are written in the standard form as $f_{\mathbf{W}}(\mathbf{x}) = \mathbf{W}_L \phi(\mathbf{W}_{L-1} \phi(\ldots \phi(\mathbf{W}_1 \mathbf{x}) \ldots))$ where $\mathbf{W}_i \in \mathbb{R}^{h_i \times h_{i-1}}$ is the weight matrix corresponding to layer $i$ of the deep network and $\phi : \mathbb{R} \to \mathbb{R}$ is a Lipschitz activation function that is applied coordinate-wise. We consider in this paper the Lipschitz constant of the activation to be $1$, which is the case for common activation functions in deep learning like ReLU and Leaky ReLU. We will use $\mathcal{F}_L$ to denote the class of spectral norm and rank bounded $L$-layer deep networks;

$$\mathcal{F}_L := \{f_{\mathbf{W}}(\mathbf{x}) = \mathbf{W}_L \phi(\mathbf{W}_{L-1} \phi(\ldots \phi(\mathbf{W}_1 \mathbf{x}) \ldots)) \mid \|\mathbf{W}_i\|_2 \leq B_i, \operatorname{rank}(\mathbf{W}_i) \leq r_i\} \quad (1)$$

This function class is different from the Frobenius norm bounded class of functions considered in Golowich et al. (2018). Working with the class of spectral norm bounded functions allows us to obtain generalization bounds that depend on the rank of the weight matrices instead of the Frobenius norm.

**Gaussian and Rademacher complexity.** Empirical Gaussian and Rademacher complexities, denoted as $\hat{\mathcal{G}}_S(\cdot)$ and $\hat{\mathcal{R}}_S(.)$ respectively, are two common measures used in statistical learning theory to compute generalization bounds. They measure the size a function class $\mathcal{F}$ by mapping it to the set of images of a sample $S = \{\mathbf{x}_1, \ldots, \mathbf{x}_m\}$, i.e., the set $\{[f(\mathbf{x}_1, \ldots f(\mathbf{x}_m)|f \in \mathcal{F}\} \subseteq \mathbb{R}^m$, and subsequently measuring its Gaussian or Rademacher width;

$$\hat{\mathcal{G}}_S(\mathcal{F}) = \frac{1}{m}\mathbb{E}_\gamma \sup_{f\in\mathcal{F}} \sum_{i=1}^m \gamma_i f(\mathbf{x}_i) \quad \text{and} \quad \hat{\mathcal{R}}_S(\mathcal{F}) = \frac{1}{m}\mathbb{E}_\sigma \sup_{f\in\mathcal{F}} \sum_{i=1}^m \sigma_i f(\mathbf{x}_i)$$

where $\gamma = \{\gamma_i\}_{i=1}^m$ is a sequence of Gaussian variables $\gamma_i \sim \mathcal{N}(0,1)$ and $\sigma = \{\sigma_i\}_{i=1}^m$ a sequence of Rademacher variables $\mathbb{P}(\sigma_i = +1) = \mathbb{P}(\sigma_i = -1) = 1/2$. The measures evaluate the capacity of a function class to fit random noise patterns, by measuring its correlation with a random sequence. While most generalization bounds are based on Rademacher complexity, the two measures are nearly interchangeable and only differ by a constant factor $\hat{\mathcal{R}}_S(\mathcal{F}) \leq \sqrt{\pi/2}\hat{\mathcal{G}}_S(\mathcal{F})$. While the empirical complexities depend on the sample $S$, we can also take an expectation wrt. the sample to obtain the true Gaussian or Rademacher complexities $\mathcal{G}_m(\mathcal{F}) = \mathbb{E}_S[\hat{\mathcal{G}}_S(\mathcal{F})]$, $\mathcal{R}_m(\mathcal{F}) = \mathbb{E}_S[\hat{\mathcal{R}}_S(\mathcal{F})]$.

**Generalization in statistical learning.** In machine learning we estimate the parameters of a model $f_\mathbf{W}$ through the minimization of a loss function $g(\mathbf{z}) = \ell(f_\mathbf{W}(\mathbf{x}), \mathbf{y})$ on a training set $S = \{(\mathbf{x}_i, \mathbf{y}_i)\}_{i=1}^m$. This training set is assumed to be drawn i.i.d from a distribution $\mathcal{D}$ on $\mathcal{X} \times \mathcal{Y}$. While we minimize the loss on the training set, our goal is to find a solution that generalizes to unseen data drawn from the same distribution. The gap between the performance of a model on new, unseen data and its performance on the training set is called the generalization error $\epsilon_{\text{gen}}$ or generalization gap of the model.

$$\epsilon_{\text{gen}} := \mathbb{E}_{\mathbf{z}\sim\mathcal{D}}\left[g(\mathbf{z})\right] - \frac{1}{m}\sum_{i=1}^m g(\mathbf{z}_i)$$

Statistical learning theory is concerned with providing guarantees on the generalization error of the models we obtain from a given training dataset and procedure. One of the classical approaches for proving generalization is the uniform convergence approach which aims to bound the generalization gap by the size of the hypothesis class to which the model belongs. Different notions of the size such as the VC dimension and Rademacher complexity have been used to provide bounds on the generalization gap for statistical learning problems Shalev-Shwartz and Ben-David (2014); Mohri et al. (2018). Since in this paper we compute the Gaussian complexity of rank-constrained deep networks, we state the following result from Bartlett and Mendelson (2002) that shows how the generalization gap of a deep network is bounded by its Gaussian complexity.

**Theorem 1 (Gaussian Complexity Generalization Bound)** *Let $\mathcal{H}$ be a function class of hypotheses composed with losses mapping from $\mathcal{Z} = \mathcal{X} \times \mathcal{Y}$ to $[0, 1]$. Then, for any $\delta > 0$, with probability at least $1 - \delta$ over the draw of an i.i.d. sample $S$ of size $m$, the following holds for all $g \in \mathcal{H}$:*

$$\mathbb{E}[g(\mathbf{z})] \leq \frac{1}{m}\sum_{i=1}^m g(\mathbf{z}_i) + \sqrt{2\pi}\hat{\mathcal{G}}_S(\mathcal{H}) + \sqrt{\frac{9\log\frac{2}{\delta}}{2m}}$$

## 3. Gaussian Complexity for Rank-Dependent Bounds

**Norm based definitions.** Prior work on generalization bounds has typically focused on deep networks where the weights layers are bounded in the Frobenius norm $||.||_F$. They usually proceed by deriving a process to "peel" the layers of a deep network one by one in order to obtain a bound on the Rademacher complexity of the whole network. For example Lemma 1 of Golowich et al. (2018) makes use of the following peeling step;

$$\mathbb{E}_\sigma \left[ \sup_{f \in \mathcal{F}; \mathbf{W}: \|\mathbf{W}\|_F \leq B} g \left( \left\| \sum_{i=1}^m \sigma_i \phi(\mathbf{W} f(\mathbf{x}_i)) \right\| \right) \right] \leq 2 \cdot \mathbb{E}_\sigma \left[ \sup_{f \in \mathcal{F}} g \left( B \left\| \sum_{i=1}^m \sigma_i f(\mathbf{x}_i) \right\| \right) \right]$$

which suggests a norm-based definition of Gaussian complexity for vector valued functions

$$\hat{\mathcal{G}}_S^{\text{norm}}(\mathcal{F}) = \mathbb{E}_\gamma \left[ \sup_{f \in \mathcal{F}} \left\| \frac{1}{m} \sum_{i=1}^m \gamma_i f(\mathbf{x}_i) \right\| \right]. \tag{2}$$

**The problem to capture the effect of rank.** This norm-based definition of Gaussian complexity however does not help us obtain generalization bounds that depend on the rank of functions. We can illustrate this with an example from the simple linear case. Consider the following classes of linear functions bounded in spectral norm, $\mathcal{F} = \{f_\mathbf{W}(\mathbf{x}) = \mathbf{W}\mathbf{x}| \|\mathbf{W}\|_2 \leq B\}$ and $\mathcal{F}_1 = \{f_\mathbf{W}(\mathbf{x}) = \mathbf{W}\mathbf{x}| \|\mathbf{W}\|_2 \leq B; \text{rank}(\mathbf{W}) = 1\}$. We can show that $\hat{\mathcal{G}}_S^{\text{norm}}(\mathcal{F}_1) = \hat{\mathcal{G}}_S^{\text{norm}}(\mathcal{F})$. To see this let us denote $\mathbf{v} = \frac{1}{m} \sum_{i=1}^m \gamma_i \mathbf{x}_i$. Then $\hat{\mathcal{G}}_S^{\text{norm}}(\mathcal{F}_1) = \mathbb{E}_\gamma \sup_{\|\mathbf{W}\|_2 \leq B} \|\mathbf{W}\mathbf{v}\|_2$. If we set $\mathbf{W} = \frac{B}{\|\mathbf{v}\|_2^2} \mathbf{v}\mathbf{v}^\top$ we find that $\mathbf{W}$ is rank-1 and maximizes $\|\mathbf{W}\mathbf{v}\|_2$. We finally obtain the bound $\hat{\mathcal{G}}_S^{\text{norm}}(\mathcal{F}_1) \leq \frac{B \max_i \|\mathbf{x}_i\|_2}{\sqrt{m}}$. We obtain a similar upper bound for $\hat{\mathcal{G}}_S^{\text{norm}}(\mathcal{F})$ even though we allow for full rank matrices. Hence this norm-based definition of Gaussian complexity for vector-valued functions does not help us distinguish between functions with different rank constraints, even though allowing for higher rank functions should mean that we have a more expressive function class.

**A suitable vector valued Gaussian complexity definition.** In this paper, we instead use the following definition of Gaussian complexity for vector valued functions, that introduces Gaussian variables for each components of the function's output representations. For a class of functions $\mathcal{F} = \{f : \mathbb{R}^d \rightarrow \mathbb{R}^k\}$;

$$\hat{\mathcal{G}}_S(\mathcal{F}) = \mathbb{E}_\gamma \left[ \sup_{f \in \mathcal{F}} \frac{1}{m} \sum_{i=1,j=1}^{m,k} \gamma_{ji} f_j(\mathbf{x}_i) \right] \tag{3}$$

where $f_j$ denotes the $j^{\text{th}}$ output coordinate of $f$. This definition appears in Maurer (2016b) and is shown to satisfy a vector contraction inequality under composition with $\mathcal{L}$-Lipschitz functions. This vector contraction inequality proves useful when bounding the composition of a class of networks with a loss function (see Theorem 2).

$$\mathbb{E} \sup_{f \in F} \sum_i \gamma_i g_i(f(\mathbf{x}_i)) \leq \sqrt{2\mathcal{L}} \mathbb{E} \sup_{f \in F} \sum_{i,j} \gamma_{ij} f_j(\mathbf{x}_i)$$

As we will show in subsequent section, this definition of Gaussian complexity also allows us to obtain generalization bounds that are sensitive to the rank of linear functions, as well as deep networks.

We will make use of the following theorem that puts together a few standard results to obtain a generalization bound for vector-valued hypothesis classes. A Rademacher complexity version of the following result is stated and proved in theorem 12 in appendix B.

**Theorem 2 (Vector-valued Gaussian complexity Generalization Bound)** *Let $\mathcal{F}$ be a family of neural networks mapping from $\mathcal{X}$ to $\mathcal{Z}$ and let $\mathcal{H}$ be a family of $\mathcal{L}$-Lipschitz functions mapping from $\mathcal{Z}$ to $[0,1]$. Then, for any $\delta > 0$, with probability at least $1 - \delta$ over the draw of an i.i.d. sample $S$ of size $m$, the following holds for all $g \in \mathcal{H}$ and $f \in \mathcal{F}$:*

$$\mathbb{E}[g(f(\mathbf{x}))] \leq \frac{1}{m} \sum_{i=1}^{m} g(f(\mathbf{x}_i)) + \sqrt{\mathcal{L}\pi}\hat{\mathcal{G}}_S(\mathcal{F}) + \sqrt{\frac{9\log\frac{2}{\delta}}{2m}}$$

In the next section we will derive bounds on the Gaussian complexity (as defined in (3)) of rank-constrained deep networks. We can plug these into the setting of theorem 2 to obtain the corresponding generalization bounds.

## 4. Gaussian Complexity Bound for Low Rank Deep Networks

Traditional approaches to obtaining bounds on the Rademacher or Gaussian complexity of deep networks typically proceed by bounding the complexity of a single layer, then sequentially "peeling" the layers to derive a complexity bound for the entire network. However, this method fails to account for interactions across the layers of a deep network. In this section, we apply the chain rule developed by Maurer (2016a) to obtain Gaussian complexity bounds for rank-constrained deep networks. We demonstrate how low-rank layers can prevent the multiplication of rank and dimension-dependent quantities across layers, leveraging this insight to establish a Gaussian complexity bound for low-rank deep networks in Theorem 7.

### 4.1. Maurer's chain rule

Our main result leverages a chain rule for the expected suprema of Gaussian processes developed by Maurer (2016a). This chain rule is key in our demonstration and serves as an alternative to Golowich et al. (2018) for "peeling" each layer of the deep neural network. Maurer's Gaussian complexity chain rule is presented as;

**Theorem 3 (Maurer Gaussian Complexity Chain Rule)** *Let $Y \subseteq \mathbb{R}^p$ be finite, $F$ a finite class of functions $f : Y \to \mathbb{R}^q$. Then there are universal constants $C_1$ and $C_2$ such that for any $\mathbf{y}_0 \in Y$*

$$G(F(Y)) \leq C_1 L(F)G(Y) + C_2 D(Y)R(F,Y) + G(F(\mathbf{y}_0))$$

where $G(\cdot)$ denotes the unnormalized Gaussian complexity, $L(F)$ is the Lipschitz constant of the set $F$, $D(Y)$ is the diameter of the set $Y$ and $R(F,Y)$ can be thought of as the Gaussian average of Lipschitz quotients of $F$. More precisely, these quantities are:

$$D(Y) = \sup_{y,y' \in Y} \|y - y'\| \quad \text{and} \quad R(F,Y) = \sup_{y,y' \in Y, y \neq y'} \mathbb{E} \sup_{f \in F} \frac{\langle \gamma, f(y) - f(y') \rangle}{\|y - y'\|}$$

In the rest of this section we will bound each of the above quantities and put them together to obtain a bound on the Gaussian complexity of deep networks. In our results we will be using normalized versions of the Gaussian complexity, which means we will divide both sides of the chain rule by the sample size $m$.

### 4.2. Gaussian complexity and Diameter of deep networks

Our insight into how low rank deep networks can avoid multiplicative factors across layers in their complexity stems from their diameter. We first illustrate this idea by computing a gaussian complexity bound for deep linear networks.

**Lemma 4 (Gaussian complexity of a deep linear network)** *Let the class of spectrally bounded deep linear network* $\mathcal{F}_L = \{f_{\mathbf{W}}(\mathbf{x}) = \mathbf{W}_L \mathbf{W}_{L-1} \ldots \mathbf{W}_1 \mathbf{x}) \mid \|\mathbf{W}_i\|_2 \leq B, \mathbf{W}_i \in \mathbb{R}^{h_i \times h_{i-1}}, h_0 = d\}$. *The Gaussian complexity of this function class is upper bounded as follow;*

$$\hat{\mathcal{G}}_S(\mathcal{F}) \leq R\sqrt{\frac{h_L \times \min_i \operatorname{rank}(\mathbf{W}_i)}{m}} \prod_{i=1}^{L} \|\mathbf{W}_i\|_2$$

*where* $h_i$ *denotes the width of hidden layer* $i$, $\mathbf{X} = [\mathbf{x}_1, \ldots, \mathbf{x}_m]^\top$, *and* $\max_i \|\mathbf{x}_i\| \leq R$.

**Proof** For single layer linear functions of width $h$, we can bound their Gaussian complexity as $R\|\mathbf{W}\|_F \sqrt{\frac{h}{m}}$ (see Lemma 10 in the appendix). We know that for rank-$r$ matrices $\|\mathbf{W}\|_F \leq \|\mathbf{W}\|_2 \times \sqrt{r}$. For deep linear networks, we can plug in $\mathbf{W} = \prod_{i=1}^{L} \mathbf{W}_i$ into the above bound. The bound follows from the fact that $\operatorname{rank}(\mathbf{W}) \leq \min_i \operatorname{rank}(\mathbf{W}_i)$ and $\|\mathbf{W}\|_2 \leq \prod_{i=1}^{L} \|\mathbf{W}_i\|_2$. ∎

This observation that the complexity is controlled by the smallest rank matrix, and not the product of the norms of the entire chain of matrices is favorable for avoiding exponential dependencies on the depth for the rank. We now exploit this fact to bound the diameter of deep nonlinear networks. This allows us to obtain favorable bounds on the intermediate diameter $D(\mathcal{F}_i(\mathbf{X}))$ for the map from the input to the $i$th layer.

**Lemma 5 (Diameter of the deep nonlinear network function class)** *Let the class of spectral norm bounded deep networks with* $\ell$ *layers be* $\mathcal{F}_\ell = \{f_{\mathbf{W}}(\mathbf{x}) = \phi(\mathbf{W}_\ell \phi(\mathbf{W}_{\ell-1}\phi(\ldots \phi(\mathbf{W}_1 \mathbf{x})\ldots))) \mid \|\mathbf{W}_i\|_2 \leq B, \mathbf{W}_i \in \mathbb{R}^{h_i \times h_{i-1}}, h_0 = d\}$, *where* $\phi$ *is a Lipschitz continuous* $(\mathcal{L}(\phi) \leq 1)$, *piecewise linear activation function (ReLU or Leaky ReLU). If* $\mathbf{X} \in \mathbb{R}^{d \times m}$ *is a given sample of data, the diameter of the function class (projected onto the data sample) can be bounded as:*

$$D(\mathcal{F}_\ell(\mathbf{X})) \leq \|\mathbf{X}\|_F \sqrt{2 \min_{1 \leq j \leq \ell} \operatorname{rank}(\mathbf{W}_j)} \times 2 \prod_{j=1}^{\ell} \|\mathbf{W}_j\|_2$$

**Proof** The diameter of the image of a data sample $\mathbf{X}$ through a function class $\mathcal{F}_\ell$ is defined as $D(\mathcal{F}_\ell(\mathbf{X})) = \sup_{f,f' \in \mathcal{F}_\ell} ||f(\mathbf{X}) - f'(\mathbf{X})||_F$.

$$D(\mathcal{F}_\ell(\mathbf{X})) = \sup_{j \le \ell: \mathbf{W}_j, \mathbf{W}'_j} ||\phi(\mathbf{W}_\ell \phi(\ldots \phi(\mathbf{W}_1 \mathbf{X}) \ldots)) - \phi(\mathbf{W}'_\ell \phi(\ldots \phi(\mathbf{W}'_1 \mathbf{X}) \ldots))||_F$$

$$\le \sup_{j \le \ell: \mathbf{W}_j, \mathbf{W}'_j} ||\mathbf{W}_\ell \phi(\ldots \phi(\mathbf{W}_1 \mathbf{X}) \ldots) - \mathbf{W}'_\ell \phi(\ldots \phi(\mathbf{W}'_1 \mathbf{X}) \ldots)||_F$$

$$= \sup_{j \le \ell: \mathbf{W}_j, \mathbf{W}'_j} \sqrt{\sum_{i=1}^m ||\mathbf{W}_\ell \phi(\ldots \phi(\mathbf{W}_1 \mathbf{x}_i) \ldots) - \mathbf{W}'_\ell \phi(\ldots \phi(\mathbf{W}'_1 \mathbf{x}_i) \ldots)||_2^2}$$

$$= \sup_{j \le \ell: \mathbf{W}_j, \mathbf{W}'_j} \sqrt{\sum_{i=1}^m ||\mathbf{W}_\ell \mathbf{D}_{\ell-1}^i \ldots \mathbf{D}_1^i \mathbf{W}_1 \mathbf{x}_i - \mathbf{W}'_\ell \mathbf{D}_{\ell-1}'^i \ldots \mathbf{D}_1'^i \mathbf{W}'_1 \mathbf{x}_i||_2^2}$$

$$\le \sup_{j \le \ell: \mathbf{W}_j, \mathbf{W}'_j} \sqrt{\sum_{i=1}^m ||\underbrace{\mathbf{W}_\ell \mathbf{D}_{\ell-1}^i \ldots \mathbf{D}_1^i \mathbf{W}_1}_{\mathbf{A}^i} - \underbrace{\mathbf{W}'_\ell \mathbf{D}_{\ell-1}'^i \ldots \mathbf{D}_1'^i \mathbf{W}'_1}_{\mathbf{A}'^i}||_F^2 ||\mathbf{x}_i||^2}$$

$$\le \sup_{j \le \ell: \mathbf{W}_j, \mathbf{W}'_j} \max_{1 \le i \le m} ||\mathbf{A}^i - \mathbf{A}'^i||_F ||\mathbf{X}||_F$$

Here $\{\mathbf{D}_j^i, \mathbf{D}_j'^i\}_{j=1}^\ell$ are diagonal matrices that correspond to the patterns of the piecewise linear 1-Lipschitz activation function $\phi$ for the two functions $f, f'$ evaluated on input $\mathbf{x}_i$.

We will now proceed to bound $\max_{1 \le i \le m} ||\mathbf{A}^i - \mathbf{A}'^i||_F$ through the rank and spectral norm of $\mathbf{A}^i, \mathbf{A}'^i$. For a class of rank-constrained, spectral-norm bounded matrices $\{\mathbf{P} | \text{rank}(\mathbf{P}) \le r, ||\mathbf{P}||_2 \le B\}$, we can bound its Frobenius norm diameter as $||\mathbf{P} - \mathbf{Q}||_F \le \sqrt{2r} \times 2B$.

Since the matrices $\mathbf{D}_j^i$ may be full rank, we have that $\forall i, \text{rank}(\mathbf{A}^i) = \text{rank}(\mathbf{A}'^i) \le \min_j \text{rank}(\mathbf{W}_j)$.

Since the entries of $\mathbf{D}_j^i$ are bounded by 1 ($\mathcal{L}(\phi) \le 1$), we also have that $\forall i, ||\mathbf{A}^i||_2 = ||\mathbf{A}'^i||_2 \le \prod_{j=1}^\ell ||\mathbf{W}_j||_2$.

Putting the above arguments together, we get the required bound on the diameter $D(\mathcal{F}_\ell(\mathbf{X})) \le ||\mathbf{X}||_F \sqrt{2 \min_j \text{rank}(\mathbf{W}_j)} \times 2 \prod_{j=1}^\ell ||\mathbf{W}_j||_2$. ∎

### 4.3. Gaussian average of Lipschitz coefficients

We now turn our attention to the next term in the chain rule – the Gaussian average of Lipschitz coefficients $R(F, Y)$. In our case, we take $F$ to be a linear transformation $\mathbf{W}_\ell$, and $Y$ to be the output of the network until the previous layer $\phi(\mathbf{W}_{\ell-1} \ldots \phi(\mathbf{W}_1 \mathbf{x}) \ldots)$. We can state the following:

**Lemma 6 (Gaussian average of Lipschitz coefficients)** *Let $\mathcal{F} = \{\mathbf{W}_\ell \mathbf{x} \mid ||\mathbf{W}_\ell||_2 \le B, \mathbf{W}_\ell \in \mathbb{R}^{h_\ell \times h_{\ell-1}}\}$ be a layer of a deep network bounded in spectral norm and $\mathcal{F}_{\ell-1} = \{f_\mathbf{W}(\mathbf{x}) = \phi(\mathbf{W}_{\ell-1} \phi(\ldots \phi(\mathbf{W}_1 \mathbf{x}) \ldots)) \mid ||\mathbf{W}_i||_2 \le B, \mathbf{W}_i \in \mathbb{R}^{h_i \times h_{i-1}}, h_0 = d\}$ be the set of functions from the input to the previous layer. Let $\mathbf{X} = [\mathbf{x}_1, \ldots, \mathbf{x}_m]^\top$ be a data sample. Then $R(\cdot, \cdot)$ can be expressed as;*

$$R(\mathcal{F}, \mathcal{F}_{\ell-1}) \le ||\mathbf{W}_\ell||_F \sqrt{h_\ell}$$

**Proof** Let us consider the projection of the function class $\mathcal{F}_{\ell-1}$ onto the data sample $\mathbf{X}$. That is, let $\mathbf{Z}_{\ell-1} = f(\mathbf{X})$ and $\mathbf{Z}'_{\ell-1} = f'(\mathbf{X})$ for $f, f' \in \mathcal{F}_{ell-1}$. From the definition of $R(\cdot, \cdot)$ we have that

$$
\begin{aligned}
R(\mathcal{F}, \mathcal{F}_{\ell-1}) &= \sup_{\mathbf{Z}_{\ell-1}, \mathbf{Z}'_{\ell-1}} \mathbb{E}_\gamma \left[ \sup_{\|\mathbf{W}_\ell\|_2 \le B} \frac{\langle \mathbf{W}_\ell, \Gamma(\mathbf{Z}_{\ell-1} - \mathbf{Z}'_{\ell-1}) \rangle}{\|\mathbf{Z}_{\ell-1} - \mathbf{Z}'_{\ell-1}\|_F} \right] \\
&\le \sup_{\mathbf{Z}_{\ell-1}, \mathbf{Z}'_{\ell-1}} \frac{\|\mathbf{W}_\ell\|_F}{\|\mathbf{Z}_{\ell-1} - \mathbf{Z}'_{\ell-1}\|_F} \mathbb{E}_\gamma \|\Gamma(\mathbf{Z}_{\ell-1} - \mathbf{Z}'_{\ell-1})\|_F
\end{aligned}
$$

In the above, $\Gamma \in \mathbb{R}^{h_\ell \times m}$ is a matrix of independent Gaussian variables.
We have that $\mathbb{E}_\gamma \left[ \|\Gamma(\mathbf{Z}_{\ell-1} - \mathbf{Z}'_{\ell-1})\|_F \right] \le \sqrt{h_\ell} \|\mathbf{Z}_{\ell-1} - \mathbf{Z}'_{\ell-1}\|_F$. Plugging this into the above inequalities we get the desired result. ∎

### 4.4. Main Result

We have introduced the chain rule that we employ, as well as our main insight into how low rank deep networks avoid multiplying factors across layers. We are now ready to state and prove our main theorem that bounds the Gaussian complexity of low rank deep networks.

**Theorem 7 (Gaussian complexity of deep Lipschitz neural network)** *Let $\mathcal{F}_L$ be the class of rank and spectral norm constrained deep neural networks of depth $L$ with a piecewise linear $1$-Lipschitz activation function $\phi$ (such as ReLU, Leaky ReLU).*

$$
\mathcal{F}_L := \{ f_{\mathbf{W}}(\mathbf{x}) = \mathbf{W}_L \phi(\mathbf{W}_{L-1} \phi(\dots \phi(\mathbf{W}_1 \mathbf{x}) \dots)) \mid \|\mathbf{W}_i\|_2 \le B_i, rank(\mathbf{W}_i) \le r_i \}
$$

*The width of layer $i$ is $h_i$, which means the dimensions of $\mathbf{W}_i$ are $h_i \times h_{i-1}$ with $h_0 = d$. Let $\mathbf{X} = [\mathbf{x}_1 \dots \mathbf{x}_m]^\top \in \mathbb{R}^{m \times d}$ denote the matrix corresponding to the data sample $S$. We assume that the data are all bounded, with $\max_i \|\mathbf{x}_i\| \le R$. The Gaussian complexity of $\mathcal{F}_L$ can be upper bounded as:*

$$
\hat{\mathcal{G}}_S(\mathcal{F}_L) \lesssim \frac{\|\mathbf{X}\|_F}{m} \left\{ \left( \|\mathbf{W}_1\|_F \sqrt{h_1} \right) \prod_{i=2}^{L} C_1 \|\mathbf{W}_i\|_2 + \sum_{i=2}^{L} C_1^{L-i} C_2 2 \sqrt{2 \mathfrak{r}_i} \kappa_i \|\mathbf{W}_i\|_F \sqrt{h_i} \right\} \quad (4)
$$

*where $\kappa_i \propto \left( \prod_{j=1, j \ne i}^{d} \|\mathbf{W}_j\|_2 \right)$ and $\mathfrak{r}_i = \min_{j \le i}(rank(\mathbf{W}_j))$.*

Equation (4) presents our bound in full generality, without using the explicit bounds on the ranks of the layers. When we further bound the Frobenius norms of each weight matrix in the above Theorem as $\|\mathbf{W}_i\|_F \le \sqrt{rank(\mathbf{W}_i)} \|\mathbf{W}_i\|_2 \le \sqrt{r} B_i$, and let $h = \max_i h_i$ be the maximum width, we can simplify the Gaussian complexity bound to $\mathcal{O}\left( C_1^L \prod_{i=1}^{L} B_i R L r \sqrt{\frac{h}{m}} \right)$.

### 4.5. Detailed proof of Theorem 7

Recalling Maurers chain rule on Gaussian processes, for any composition $F \circ Y$ we can write;

$$
\frac{1}{m} G(F \circ Y) \le \frac{C_1}{m} L(F) G(Y) + \frac{C_2}{m} D(Y) R(F, Y) + \frac{1}{m} G(F(\mathbf{y}_0))
$$

where $D(\cdot)$ is the diameter of the function class and $R(\cdot, \cdot)$ is the average of Lipschitz coefficients. We can omit the third term in the chain rule since we can choose $\mathbf{y}_0$ to be the zero function (which can be obtained by setting the weight matrices to zero). WLOG we can drop the last term in the chain rule. Let us define the image of the data up to layer $i$ as $\mathbf{Z}_i = f(\mathbf{X})$ for $f \in \mathcal{F}_i$ (similar to Lemma 6).

**The base recursion of the chain.** Given above notation, we observe $\hat{\mathcal{G}}_S(\mathbf{Z}_i) \leq \mathcal{L}(\phi)\hat{\mathcal{G}}_S((\mathcal{F}_{i-1})$. This is a direct application of Talagrand's contraction lemma. We can use this simple observation to write the recursion;

$$\hat{\mathcal{G}}_S(\mathcal{F}_i) = C_1 \mathcal{L}(\phi) ||\mathbf{W}_i||_2 \hat{\mathcal{G}}_S(\mathcal{F}_{i-1}) + C_2 \frac{1}{m} D(\mathbf{Z}_{i-1}) R(\mathbf{W}_i, \mathbf{Z}_{i-1}) \quad \forall i > 1$$

$$\hat{\mathcal{G}}_S(\mathcal{F}_1) \leq ||\mathbf{W}_1||_F ||\mathbf{X}||_F \frac{\sqrt{h_1}}{m} \quad \text{Complexity of first layer pre-activation.}$$

In this recursion, we also used the fact that the Lipschitz coefficient of a linear transformation is its spectral (or operator) norm measuring the maximum spread of the matrix; $L(\mathbf{W}_i) = ||\mathbf{W}_i||_2$.

**Getting to a closed form.** Following induction until the end of the chain gives us, under the convention that a product over an empty set is 1 and the sum over an empty set is 0:

$$\hat{\mathcal{G}}_S(\mathcal{F}_L) = \underbrace{\left( \prod_{i=2}^{L} C_1 \mathcal{L}(\phi) ||\mathbf{W}_i||_2 \right) \hat{\mathcal{G}}_S(\mathcal{F}_1)}_{\text{direct recursive contribution}}$$

$$+ \underbrace{\sum_{i=2}^{L} \left( \left( \prod_{j=i+1}^{L} C_1 \mathcal{L}(\phi) ||\mathbf{W}_j||_2 \right) C_2 \frac{1}{m} D(\mathbf{Z}_{i-1}) R(\mathbf{W}_i, \mathbf{Z}_{i-1}) \right)}_{\text{sum of disturbance contributions}}.$$

This closed form formula consists of all the directive recursive contributions occurring from the first term in the chain rule $C_1 L(F)G(Y)$ plus a a contribution that is a sum of disturbance for every additional layer we peel that comes from the second term in the chain rule $C_2 D(Y)R(F, Y)$. We have bounded each of the terms in the above recursion in the prior subsections. $\hat{\mathcal{G}}_S(\mathcal{F}_1)$ follows from standard computations (see lemma 10 in appendix), $D(\mathbf{Z}_{i-1})$ is computed in lemma 5 and $R(\mathbf{W}_i, \mathbf{Z}_{i-1})$ is computed in lemma 6.

**Putting it all together.** Using the bounds computed in the prior subsections, and using $\mathcal{L}(\phi) \leq 1$ we can now stitch everything together to get the bound of our Theorem (ignoring constants):

$$\hat{\mathcal{G}}_S(\mathcal{F}_L) \lesssim \frac{||\mathbf{X}||_F}{m} \left\{ \left( ||\mathbf{W}_1||_F \sqrt{h_1} \right) \prod_{i=2}^{L} C_1 ||\mathbf{W}_i||_2 + \sum_{i=2}^{L} C_1^{L-i} C_2 2\sqrt{2\mathfrak{r}_i} \kappa_i ||\mathbf{W}_i||_F \sqrt{h_i} \right\} \quad (5)$$

where $\kappa_i \propto \left( \prod_{j=1, j \neq i}^{d} ||\mathbf{W}_j||_2 \right)$ and $\mathfrak{r}_i = \min_{j \leq i}(\text{rank}(\mathbf{W}_j))$. Now if we use the fact that we have spectral norm bounded networks with a maximum layer rank $\leq r$ and layer width $\leq h$, and that the data sample is bounded as $||\mathbf{x}_i||_2 \leq R$, our bound simplifies to $\hat{\mathcal{G}}_S(\mathcal{F}_L) \lesssim C_1^L \prod_{i=1}^{L} ||\mathbf{W}_i||_2 RLr \sqrt{\frac{h}{m}}$

## 5. Discussion

**Comparison with existing bounds.**    In the previous section we used a chain rule due to Maurer to obtain Gaussian complexity bounds for rank-constrained deep networks. We now compare our result to generalization bounds previously obtained in the literature. We present our bounds as well as the bounds we compare to in Table 1. We find that our bounds compare favorably to existing bounds when we evaluate them on the rank and spectral norm constrained class of deep networks. While our bound contains an unfavorable factor $C_1^L$ we believe this is an artifact of the current proof technique, and may not be truly necessary. We focus our comparison to the other factors in our bound.

| Source | Original Bound | Bound for networks in $\mathcal{F}_L$ |
|---|---|---|
| Ours (2024) | $\mathcal{O}\left(\frac{\prod_{i=1}^L \|\mathbf{W}_i\|_2 C_1^L L r \sqrt{h}}{\sqrt{m}}\right)$ | $\mathcal{O}\left(\prod_{i=1}^L \|\mathbf{W}_i\|_2 \times \frac{C_1^L L r \sqrt{h}}{\sqrt{m}}\right)$ |
| Golowich et al. (2018) | $\mathcal{O}\left(\frac{\sqrt{L}\prod_{i=1}^L \|\mathbf{W}_i\|_F}{\sqrt{m}}\right)$ | $\mathcal{O}\left(\prod_{i=1}^L \|\mathbf{W}_i\|_2 \times \sqrt{\frac{L r^L}{m}}\right)$ |
| Neyshabur et al. (2018) | $\mathcal{O}\left(\prod_{i=1}^L \|\mathbf{W}_i\|_2 \times \sqrt{\frac{L^2 h \sum_{i=1}^L \frac{\|\mathbf{W}_i\|_F^2}{\|\mathbf{W}_i\|_2^2}}{m}}\right)$ | $\mathcal{O}\left(\prod_{i=1}^L \|\mathbf{W}_i\|_2 \times \sqrt{\frac{L^3 r h}{m}}\right)$ |
| Bartlett et al. (2017) | $\mathcal{O}\left(\prod_{i=1}^L \|\mathbf{W}_i\|_2 \times \frac{\left(\sum_{i=1}^L \left(\frac{\|\mathbf{W}_i^\top\|_{2,1}}{\|\mathbf{W}\|_2}\right)^{2/3}\right)^{3/2}}{\sqrt{m}}\right)$ | $\mathcal{O}\left(\prod_{i=1}^L \|\mathbf{W}_i\|_2 \times \sqrt{\frac{L^3 r h}{m}}\right)$ |

Table 1: Comparison of our results with other complexity bounds for deep networks. For each bound we present both the original formulation and its evaluation on the spectral norm and rank bounded deep network class $\mathcal{F}_L$. We denote the maximum rank of a network layer by $r$, the maximum width of a layer by $h$ and the depth of the network by $L$. We drop the factor related to the maximum norm of the input data. This can be assumed to be a constant.

First we compare our bound to that of Golowich et al. (2018) who obtain norm based generalization bounds through a careful analysis of Rademacher complexity. While their bound only has a mild explicit dependence on the depth $L$, it involves the product of Frobenius norms of the layers, which when evaluated on networks in $\mathcal{F}_L$, results in a bound that scales as $\sqrt{L} r^{L/2}$. In constrast our bound only scales as $L r \sqrt{h}$ (where $h$ is the width of the layers).

Bartlett et al. (2017) obtain generalization bounds for deep networks by upper bounding the Rademacher complexity using covering numbers. This leads to a bound that depends on the $\|\cdot\|_{2,1}$ norms of the layers. Neyshabur et al. (2018) obtain a version of this bound through a PAC-Bayesian analysis. Both of these bounds, when evaluated on our function class $\mathcal{F}_L$ scale as $\sqrt{L^3 r h}$. Our bound which scales as $\sqrt{L^2 r^2 h}$ may be better for deeper networks that also find low rank solutions. This tradeoff between depth and rank may be an interesting avenue for future research.

Our main point of contrast is the following. While complexity bounds for low rank deep networks are typically obtained by plugging in low rank matrices in a norm based complexity bound, our approach of applying Maurer's chain rule, and showing that the diameter of the chain of deep networks only

depends on the rank of the smallest rank matrix, allows us to avoid the rank factor multiplying across layers, and obtain comparable if not better generalization bounds.

**Neural Collapse improves the generalization bound.**    It was recently observed in experiments that deep classifiers become dramatically simple in the top layers in a phenomenon called Neural Collapse Papyan et al. (2020). The representations of examples belonging to the same class concentrate around the mean vector, the mean vectors spread apart to form a simplex Equiangular Tight Frame (ETF), the weight matrices and mean feature vectors become duals of each other, and the decision of the deep network follows the nearest class center classification rule. More recently, it was also observed that collapse occurs not just in the last layer of a deep network, but in intermediate layers as well Rangamani et al. (2023). Neural collapse in intermediate layers also leads to low rank layers— in fact the rank becomes precisely $C - 1$ where $C$ is the number of classes.

This simplification in the description of the top layers of a deep network must surely provide some benefit in the form of tighter generalization bounds. That is indeed the case. As a consequence of Maurer's chain rule, we can see that rank-1 layers ensure that all subsequent layers simply belong to a simple $\mathbb{R} \to \mathbb{R}$ mapping that is composed with the bottom layers of the deep network. This leads to the following bound on Rademacher complexity for networks with a rank-1 layer in the middle.

**Theorem 8 (Theorem 4 of Golowich et al. (2018))** *Let $\mathcal{G}$ be a class of functions from $\mathbb{R}^d$ to $[-R, R]$. Let $\mathcal{F}_{\ell,a}$ be the class of of $\ell$-Lipschitz functions from $[-R, R]$ to $\mathbb{R}$, such that $f(0) = a$ for some fixed $a$. Letting $\mathcal{F}_{\ell,a} \circ \mathcal{G} := \{f(g(\cdot)) : f \in \mathcal{F}_{\ell,a}, h \in \mathcal{H}\}$, its Rademacher complexity satisfies*

$$\hat{\mathcal{R}}_S(\mathcal{F}_{\ell,a} \circ \mathcal{H}) \ \leq \ c\ell \left( \frac{R}{\sqrt{m}} + \log^{3/2}(m) \cdot \hat{\mathcal{R}}_S(\mathcal{H}) \right) \ ,$$

*where $c > 0$ is a universal constant.*

This means that neural collapse in the intermediate layers of a deep network will essentially drop all of the collapsed layers from the generalization bound. The collapsed layers can be collected into the $\mathbb{R} \to \mathbb{R}$ mapping $\mathcal{F}_{\ell,a}$, and the Rademacher/Gaussian complexity of the entire network only depends on the network mapping from the input to the layer just below the collapsed layers. The depth is effectively decreased, and one can achieve a tighter generalization bound for deep networks that exhibit neural collapse.

## 6. Conclusion

In this paper we explore how low rank layers in deep networks affect their generalization beyond norm-based generalization bounds. We apply Maurer (2016a)'s chain rule to obtain gaussian complexity bounds for deep networks with low rank layers where rank factors do not multiply across layers. Our results also point to a possible rank-depth tradeoff for generalization bounds, that may be of further interest. We also identify how networks that show neural collapse have a smaller complexity than networks that do not show neural collapse, and suggest that neural collapse in intermediate layers is favorable for generalization. Studying how other types of alignment across layers can yield better generalization bounds for deep networks is another promising direction for future work.

**Limitations.** Our current results contain an undesirable exponential dependence on depth $C_1^L$ which we conjecture is an artifact of the proof technique rather than a fundamental flaw. An analogous situation is the case of norm based generalization bounds from Neyshabur et al. (2015) that contained a factor with exponential dependence on depth $(2^L)$ which was subsequently removed in later work.

## Acknowledgments

We thank Lorenzo Rosasco, Tomer Galanti, and Pierfrancesco Beneventano for many relevant discussions. This material is based on the work supported by the Center for Minds, Brains and Machines (CBMM) at MIT funded by NSF STC award CCF-1231216. Andrea Pinto was also funded by Fulbright Scholarship.

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

## Appendix A. Rank-dependent separation for one-layer networks

We first recall Talagrand's contraction lemma that helps us obtain gaussian complexity bounds for the composition of a function with a fixed $\mathcal{L}$-Lipschitz function $\phi$.

**Lemma 9 (Talagrand's Contraction Lemma)** *Let $\phi : \mathbb{R} \mapsto \mathbb{R}$ be an $\mathcal{L}$-Lipschitz function, then $\hat{\mathcal{G}}_S(\phi \circ \mathcal{H}) \leq \mathcal{L} \times \hat{\mathcal{G}}_S(\mathcal{H})$ where $\phi \circ \mathcal{H} = \{z \mapsto \phi(h(z)) : h \in \mathcal{H}\}$.*

We now present for the sake of completeness a straightforward derivation of the Gaussian complexity of a shallow low rank nonlinear network.

**Lemma 10 (Gaussian complexity of a shallow Lipschitz neural network)** *Let the class of spectrally bounded shallow Lipschitz network $\mathcal{F} = \{f_{\mathbf{W}}(\mathbf{x}) = \phi(\mathbf{W}\mathbf{x}) \mid \|\mathbf{W}\|_2 \leq B, \mathbf{W} \in \mathbb{R}^{h \times d}\}$. The Gaussian complexity of this function class is upper bounded as follow;*

$$\hat{\mathcal{G}}_S(\mathcal{F}) \leq \mathcal{L}(\phi)R\sqrt{\frac{h \times \operatorname{rank}(\mathbf{W})}{m}}\|\mathbf{W}\|_2$$

*where $h$ denotes the width of the weight matrix $\mathbf{W}$, $\mathbf{X} = [\mathbf{x}_1, \ldots, \mathbf{x}_m]^\top$, and $\max_i \|\mathbf{x}_i\| \leq R$.*

**Proof** We can write $\hat{\mathcal{G}}_S(\mathcal{F}) = (1/m)\mathbb{E}_\gamma \sup_{\mathbf{W}} \langle \mathbf{W}, \Gamma\mathbf{X} \rangle$ where $\Gamma$ is a matrix containing the Gaussian variables of our vector-valued Gaussian complexity definition in equation (3). By Cauchy-Schwartz, we get that

$$\sup_{\mathbf{W}} \langle \mathbf{W}, \Gamma\mathbf{X} \rangle \leq \|\mathbf{W}\|_F \times \|\Gamma\mathbf{X}\|_F \leq \sqrt{\operatorname{rank}(\mathbf{W})}\|\mathbf{W}\|_2\|\Gamma\mathbf{X}\|_F$$

By using the fact that $\mathbb{E}|Y| \leq \sqrt{\mathbb{E}Y^2}$ for any r.v. $Y$ we get:

$$\mathbb{E}_\gamma\left[\|\Gamma\mathbf{X}\|_F\right] \leq \sqrt{\mathbb{E}_\gamma\left[\|\Gamma\mathbf{X}\|_F^2\right]} = \sqrt{\mathbb{E}_\gamma\left[\operatorname{Tr}\left(\mathbf{X}^\top\Gamma^\top\Gamma\mathbf{X}\right)\right]} = \sqrt{\operatorname{Tr}\left(\mathbf{X}\mathbf{X}^\top\mathbb{E}_\gamma\left[\Gamma^\top\Gamma\right]\right)}$$

$$= \sqrt{\operatorname{Tr}\left(\mathbf{X}\mathbf{X}^\top \times h\mathbf{I}_m\right)} \leq R\sqrt{hm}$$

The bound follows by applying Talagrand's contraction lemma on top of the above bound. ∎

In particular, for full-rank matrices with bounded spectral norm we obtain that the Gaussian complexity $\hat{\mathcal{G}}_S(\mathcal{F}) \leq \mathcal{L}(\phi)BR\sqrt{hd^*/m}$ where $d^* = \min(d, h)$, whereas the low-rank bound leads to the bound $\mathcal{L}(\phi)BR\sqrt{hr/m}$ with $r \leq d^*$. This shows that the Gaussian complexity of one layer of rank-$r$ linear functions is smaller than that of one layer full rank ($d^*$) functions by a factor of $\sqrt{r/d^*}$.

## Appendix B. Rademacher complexity of vector-valued functions

**Theorem 11 (Rademacher Generalization Bound.)** *Let $\mathcal{G}$ be a family of functions mapping from $\mathcal{Z}$ to $[0, 1]$. Then, for any $\delta > 0$, with probability at least $1 - \delta$ over the draw of an i.i.d. sample $S$ of size $m$, each of the following holds for all $g \in \mathcal{G}$:*

$$\mathbb{E}[g(z)] \leq \frac{1}{m}\sum_{i=1}^m g(z_i) + 2\mathcal{R}_m(\mathcal{G}) + \sqrt{\frac{\log\frac{1}{\delta}}{2m}}$$

$$\mathbb{E}[g(z)] \leq \frac{1}{m}\sum_{i=1}^m g(z_i) + 2\hat{\mathcal{R}}_S(\mathcal{G}) + 3\sqrt{\frac{\log\frac{2}{\delta}}{2m}}$$

**Theorem 12 (Vector-valued contraction on Rademacher generalization bound)** *Let $\mathcal{G}$ be a family of Lipschitz functions mapping from $\mathcal{Z}$ to $[0,1]$ with Lipschitz constant $L$ and let $\mathcal{F}$ be a family of neural networks mapping from $\mathcal{X}$ to $\mathcal{Z}$. Then, for any $\delta > 0$, with probability at least $1 - \delta$ over the draw of an i.i.d. sample $S$ of size $m$, each of the following holds for all $g \in \mathcal{G}$ and $f \in \mathcal{F}$:*

$$\mathbb{E}[g(f(x))] \le \frac{1}{m} \sum_{i=1}^{m} g(f(x_i)) + 2\sqrt{2L}\hat{\mathcal{R}}_S(\mathcal{F}) + 3\sqrt{\frac{\log \frac{2}{\delta}}{2m}}$$

**Proof** From Theorem 11 we have that;

$$\mathbb{E}[g(f(x))] \le \frac{1}{m} \sum_{i=1}^{m} g(f(x_i)) + 2\hat{\mathcal{R}}_S(\mathcal{G}) + 3\sqrt{\frac{\log \frac{2}{\delta}}{2m}}$$

Using Maurer's result Maurer (2016b) on vector-valued Rademacher contraction we can write;

$$\begin{aligned}
\mathbb{E}[g(f(x))] &\le \frac{1}{m} \sum_{i=1}^{m} g(f(x_i)) + 2\hat{\mathcal{R}}_S(\mathcal{G}) + 3\sqrt{\frac{\log \frac{2}{\delta}}{2m}} \\
&= \frac{1}{m} \sum_{i=1}^{m} g(f(x_i)) + \frac{2}{m}\mathbb{E}_\sigma \left[ \sup_{g \circ f} \sum_i \sigma_i g(f(x_i)) \right] + 3\sqrt{\frac{\log \frac{2}{\delta}}{2m}} \\
&\le \frac{1}{m} \sum_{i=1}^{m} g(f(x_i)) + \frac{2}{m}\sqrt{2L}\mathbb{E}_\sigma \left[ \sup_f \sum_{i,k} \sigma_{ik} f_k(x_i) \right] + 3\sqrt{\frac{\log \frac{2}{\delta}}{2m}} \\
&= \frac{1}{m} \sum_{i=1}^{m} g(f(x_i)) + 2\sqrt{2L}\hat{\mathcal{R}}_S(\mathcal{F}) + 3\sqrt{\frac{\log \frac{2}{\delta}}{2m}}
\end{aligned}$$

■

Recent advances on Rademacher complexities derived a theorem for the chain rule:

**Theorem 13 (Rademacher Chain Rule Chu and Raginsky (2023))** *Let a countable [1] class $\mathcal{F}$ of functions $f : \mathbb{R}^k \to \mathbb{R}$ and a countable set $T \subset \mathbb{R}^{k \times n}$ be given. Assume that all $f \in \mathcal{F}$ are uniformly Lipschitz with respect to the Euclidean norm i.e. $\sup_{f \in \mathcal{F}} \|f\|_{Lip} \le L < \infty$. Assume also that $\mathcal{R}(T) < \infty$ and $\mathcal{R}(\mathcal{F}(T)) < \infty$. Then there exists a set $S \subset \mathbb{R}^{k \times n}$ with $\mathcal{G}(S) \le \mathcal{R}(T)$ and $\|S\|_{\infty,\infty} \le \|T\|_{\infty,\infty}$ such that*

$$\mathcal{R}(\mathcal{F}(T)) \le L_{\mathcal{F}}\mathcal{R}(T) + \Delta_2(S)\tau^b(\mathcal{F}, S) + \mathcal{R}(\mathcal{F}(t))$$

*for any $t \in S$, where $\Delta_2(S)$ is the diameter of $S$ with respect to the Frobenius norm, and where*

$$\tau(\mathcal{F}, S) := \sup_{s,t \in S} \frac{\mathbb{E}\left[\sup_{f \in \mathcal{F}} \sum_{i=1}^{n} \sigma_i |f(s_i) - f(t_i)|\right]}{\|s - t\|_2}$$

---

1. Attention is restricted to countable classes in order to avoid dealing with various measure-theoretic technicalities which can be handled in a standard manner by eg. imposing separability assumption on the relevant classes and processes.

