# OpenReview forum: "On Generalization Bounds for Neural Networks with Low Rank Layers"
_algorithmiclearningtheory.org/ALT/2025/Conference — ALT 2025_

### Official Review · Reviewer_oJ2Z · 2024-11-11
**Alternative derivation of generalization bounds using Maurer Gaussian Complexity Chain Rule, possibly improving existing results only when the depth of the network is larger than the rank of the weight matrices.**

**Rating:** 5
**Confidence:** 3

**Review:**

**Main review**
This paper uses Gaussian complexity to derive generalization bounds that are rank dependent. Specifically, the authors first consider the per-layer Gaussian complexity and use Maurer's Chain rule to analyze it. The derived bound is on the Gaussian complexity is $O(\dfrac{\sqrt{h}\sqrt{r}}{\sqrt{m}}(\prod_{i=1}^LC_1^{(i)}\|W_i\|^2 + \sum_{i=1}^LC_1^{(L-i)}C_2^{(i)}\sqrt{r}\prod_{i=1}^L\|W_i\|^2 ))$, where $h$ is the width, $L$ the depth $C_1^{(i)},C_2^{(i)}$ are constants for each layer $i$.



The alternative analysis through Maurer's Chain rule is interesting. However, it is unclear whether the derived results improve upon prior work even when $L>r$. Looking at the equation above, we can see that if $C_1, C_2$ are equal, for example, to 2  the derived bound becomes $O(\dfrac{\sqrt{h}\sqrt{r}}{\sqrt{m}}(2^L+L\sqrt{r})\prod_{i=1}^L\|W_i\|^2)$, which is always larger than $O(\sqrt{\dfrac{L^3rh}{m}}\prod_{i=1}^L\|W_i\|^2)$.  According to [1] "The constants $C_1$ and $C_2$ as they result from the proof are rather large,
because they accumulate the constants of Talagrand’s majorizing measure theorem and generic chaining", which results in the contribution of this work being limited.

[Above I rewrote eq. 5, which is used to derive the bound on the Gaussian complexity (please correct me if I missed anything).]


**Questions**

1. Could the authors clarify how they derive Theorem 2?
2. Regarding the constant $C_1^L$, which is exactly this constant in the last sentence before sec. 4.5?
3. In eq. 5 should $C_2, C_1$ have the indicators $i$ as well?

[1]:Maurer, Andreas. "A chain rule for the expected suprema of gaussian processes." Theoretical Computer Science 650 (2016): 109-122.

**Paper Award:**

No

---

> ### Author Response · Authors · 2024-11-21
>
> Thank you for your review. We have responded to questions 2,3 and your main review here - https://openreview.net/forum?id=TAvypH5yl5&noteId=la35U3YiwG
>
> Our response to question 1 in your review is here - https://openreview.net/forum?id=TAvypH5yl5&noteId=xWa47tZ0AN

---

### Official Review · Reviewer_XzyW · 2024-11-13
**Nice result on Gaussian complexity of multilayer NNs**

**Rating:** 6
**Confidence:** 4

**Review:**

The paper revisits the problem of obtaining generalization bounds for neural networks with non-linear Lipschitz activation functions at each layer. Specifically, by analyzing the Gaussian complexity of a network, they show that for deep multi-layer networks, the generalization error is bounded by a term that involves the minimum of the ranks of the weight matrices at each layer (plus other terms like the product of appropriate spectral norms).

The analysis proceeds by using a chain rule expression for Gaussian complexity that was developed in the work [Maurer 2016]. This lets them bound the Gaussian complexity by the product of the _spectral norms_ of the weight matrices, times the minimum rank. Compared to previous work that has the product of Frobenius norms (e.g., due to Golovich et al.), this bound can be significantly better as it avoids a "product of the ranks" type term.

The proof itself is quite clean: it follows from the chain rule, along with bounds on (a) the diameter of the images of inputs, and (b) the term R() that arises in the chain rule, that is related to the coefficients.

Overall, the result is nice in that it can significantly improve the known generalization bounds for rank-constrained networks (which are well-motivated in applications). The weakness is that the paper appears to be a fairly simple consequence of the chain rule for Gaussian complexity. I lean towards accepting the paper overall.

**Paper Award:**

No

---

> ### Author Response · Authors · 2024-11-21
>
> Thank you for your review. We have responded to your comment here - https://openreview.net/forum?id=TAvypH5yl5&noteId=b4uqcXWQmU

---

### Official Review · Reviewer_gQPE · 2024-11-14
**Good paper**

**Rating:** 8
**Confidence:** 4

**Review:**

This paper derives generalisation bounds for deep neural networks using the fact that weight matrices exhibit low-rank bias, and utilising a novel approach based on Gaussian complexity as the mathematical toolkit. In particular, the authors derive the Gaussian complexity of rank-constrained deep Lipschitz networks and use it to derive a generalisation bound. The bound is then compared to existing norm-based bounds (typically based on Rademacher complexity) and the rates are better/on par. Using this bound, they are able to show that neural collapse in the intermediate layers may be favourable in terms of generalisation.

Using low-rank bias and neural collapse to improve complexity-based generalisation bounds is, to my knowledge, a novel research direction and it is definitely interesting as it is in line with the current understanding of how neural networks perform. Gaussian complexity also seems like a promising new mathematical toolkit to derive generalisation bounds.

Strengths:
- The low-rank bias in generalisation bounds is underexploited and this is one of the first results in this direction.
-Gaussian complexity as a tool to derive generalisation bounds is also underexploited, and the paper is able to derive rates on par with/better than existing toolkits with this new approach.

Weaknesses:
- There is a new factor $C_1^L$ which appears in the generalisation bounds, which is not present in norm-based bounds (but the rates with respect to other quantities, the depth L and the maximum rank r, are better than these bounds).
- The bounds are not algorithm-dependent whereas low-rank bias is algorithm-dependent (but this is a shortcoming of all complexity-based bounds not only Gaussian complexity).

Questions:
-In the paper you say that you believe that the factor $C^L_1$ is an artifact of the proof and can possibly be dropped. Could you provide more intuition or explain in more details how you think the factor can be dropped?

**Paper Award:**

No

---

> ### Author Response · Authors · 2024-11-21
>
> Thank you for your review. We have responded to your comments here - https://openreview.net/forum?id=TAvypH5yl5&noteId=HgS43nbmil

---

### Official Review · Reviewer_sd6A · 2024-11-18
**Interesting topic; Paper not yet ready to be published**

**Rating:** 6
**Confidence:** 3

**Review:**

The paper is concerned with generalisation bounds for (deep) neural networks under the assumption that the weight matrices of the individual layers are assumed to be low-rank. The paper derives upper bounds on the Gaussian complexity of these functions. A key proof ingredient is Maurer's chain rule. The results of this paper extend and go significantly beyond existing work, see Table 1. Moreover, I believe that this is indeed an interesting topic to study.

However, my major concern with this paper is that it is not yet ready to be published. This has to do with the style of how many theorems and lemmas are written up. For example, take Theorem 7, equation (4). The right-hand side depends on ||X||_F and ||W_i||_2, ||W_i||_F. These are quantities which are not really defined here since the left-hand side is a supremum over all possible X and W_i. Clearly this can be fixed by replacing the quantities on the right-hand side by uniform upper bounds (which are indeed an assumption of the theorem). However, this is not only the case in Theorem 7. Namely, the same issue appears also in Lemma 6, Lemma 5, Lemma 4,
While these issues can clearly be fixed by the authors, this is a "major revision" and I feel one should not give the paper a pass in the current state.

Further comments:
1.Equation (2): "." missing at the end
2. page 6: ||W||_2 = \prod_{I=1}^n ||W_i||_2 is not true in general (only less or equal)
3. page 6: "Where Phi is ... linear activation function (ReLU or LeakyReLU)" (incomplete sentence)


-----------------------------------
I have read the other reviews and the rebuttals by the authors. I do not really agree with response. In my opinion, this is not really the way to present the main theorems, regardless of how other papers are presenting the results. However, I think one can understand what is meant by statements. For this reason, if the other reviewers are fine with it, I would not ponder on this point further.

Regarding the content of the paper, I believe that the paper should is above the acceptance threshold. The result itself is novel and interesting although the proof itself is rather straightforward.

**Paper Award:**

No

---

> ### Author Response · Authors · 2024-11-21
>
> Thank you for your review. We have responded to your comments here - https://openreview.net/forum?id=TAvypH5yl5&noteId=nM5LVfZzpv

---

### Author Rebuttal · Authors · 2024-11-21

**Response to Reviewer sd6A:**

We thank the reviewer for their positive comments about the topic being interesting. We also thank them for pointing out the typos and grammatical errors. We will make sure to correct those errors.

We understand the reviewer’s concerns about bounds such as in eq (4) containing quantities such as $||W_i||_2$. However this is in line with how results are generally presented in previous papers on generalization bounds using Rademacher/Gaussian complexity. Well cited prior works [1,2,3] all present their generalization bounds in this fashion.

The rationale behind this is that it allows the reader to understand which properties of the deep neural network control the complexity at first glance. It also allows for post-hoc evaluation of these generalization bounds as one can plug in the relevant quantities from the deep network into the complexity bound to evaluate them. Moreover, we also present our result in terms of the supremum over $||X||_F, ||W_i||_2$ in the line above section 4.5. We hope the reviewer will take this common practice into consideration in their evaluation of our paper.

[1] Bartlett, P. L., Foster, D. J., & Telgarsky, M. J. (2017). Spectrally-normalized margin bounds for neural networks. Advances in neural information processing systems, 30.

[2] Golowich, N., Rakhlin, A., & Shamir, O. (2018, July). Size-independent sample complexity of neural networks. In Conference On Learning Theory (pp. 297-299). PMLR.

[3] Neyshabur, B., Bhojanapalli, S., & Srebro, N. (2017). A pac-bayesian approach to spectrally-normalized margin bounds for neural networks. arXiv preprint arXiv:1707.09564.

---

### Author Rebuttal · Authors · 2024-11-21

**Response to Reviewer gQPE:**

We thank the reviewer for their positive feedback about our paper. We agree that low rank biases in deep networks have been underexploited in generalization bounds. Our approach shows how low rank layers can be used beyond just leading to smaller norms of weight matrices.

The unfavorable $C_1^L$ factor in our bounds arise from the universal constant $C_1$ in Maurer’s chain rule. This factor only appears because our results follow from applying that chain rule to the case of deep networks with low rank layers. A similar $2^L$ factor appears in prior norm-based generalization bounds [1] that comes from repeated application of Talagrand’s lemma to peel nonlinearities in deep networks. This factor was removed in subsequent work [2] by performing the peeling of layers in the exponent (in an approach similar to Chernoff bounds) which allowed Golowich et al. to obtain bounds that were polynomial in the depth, which were subsequently made depth independent. If we can derive a chain rule that is in the exponent, we believe that we can also get rid of our unfavorable $C_1^L$ factor.

We agree that bounds that are based on Rademacher/Gaussian complexity are algorithm independent, but these bounds can still be applied to the solutions obtained by specific training algorithms. We believe these complexity based bounds can still lead to insights about which properties of deep networks should be controlled by training algorithms to obtain solutions that generalize well.

[1] Neyshabur, B., Tomioka, R., & Srebro, N. (2015, June). Norm-based capacity control in neural networks. In Conference on learning theory (pp. 1376-1401). PMLR.

[2] Golowich, N., Rakhlin, A., & Shamir, O. (2018, July). Size-independent sample complexity of neural networks. In Conference On Learning Theory (pp. 297-299). PMLR.

---

### Author Rebuttal · Authors · 2024-11-21

**Response to Reviewer XzyW:**

We thank the reviewer for their positive comments about the paper. Indeed we acknowledge in the paper that our contribution is the application of Maurer’s chain rule to obtain Gaussian complexity bounds for deep networks with low rank layers. However we believe this leads to insights into how low rank layers can constrain the complexity beyond just reducing the norms of the deep network layer weights. Our result in lemma 5 about how low rank layers constrain the diameter of networks differentiates our work from previous norm based generalization bounds for deep networks.

---

### Author Rebuttal · Authors · 2024-11-21

**Response to Reviewer oJ2Z:**

**Response to Q3, Q2, main review:**
We thank the reviewer for their comments and apologize for the confusion that has arisen due to a slip in our notation on page 9. We use the superscript $i,j$ for $C_1, C_2$ when describing the base recursion and putting it together to obtain a formula, but these are universal constants that do not depend on the layer. $C_1,C_2$ arise from Talagrand’s majorizing measure theorem and generic chaining results [1], as the reviewer notes. Thus we do not index $C_1$ and $C_2$ in eq 5. This means $L-i$ in $C_1^{L-i}$ in eq 5 is an exponent and not an index. This also applies to $C_1^L$ in the last sentence before section 4.5. That constant is $C_1$ raised to the power $L$ (depth). We realize how our notation has led to this confusion and will make sure to correct this.

With this clarification, we agree that our bounds contain an unfavorable factor $C_1^L$ that does not strictly improve over prior results. We have acknowledged this at the beginning of section 5. However when focusing on the other factors in the bound (rank, width, depth), our results are comparable or improve upon prior results as we have described in section 5.

We believe the factor of $C_1^L$ is an artifact of the proof technique (using Maurer’s chain rule) and not a fundamental limit of the complexity of deep networks with low rank layers. Indeed a similar $2^L$ factor appears in prior norm-based generalization bounds [2] due to repeated application of Talagrand’s lemma. This was removed in [3] by peeling the layers in the exponent to obtain bounds that were polynomial in the depth, which were subsequently made depth independent. We aim to remove the $C_1^L$ factor by deriving a chain rule that is in the exponent.

[1] Maurer, A. (2016). A chain rule for the expected suprema of gaussian processes. Theoretical Computer Science, 650, 109-122.

[2] Neyshabur, B., Tomioka, R., & Srebro, N. (2015, June). Norm-based capacity control in neural networks. In Conference on learning theory (pp. 1376-1401). PMLR.

[3] Golowich, N., Rakhlin, A., & Shamir, O. (2018, July). Size-independent sample complexity of neural networks. In Conference On Learning Theory (pp. 297-299). PMLR.

[4] Maurer, A. (2016). A vector-contraction inequality for rademacher complexities. In Algorithmic Learning Theory: 27th International Conference, ALT 2016, Bari, Italy, October 19-21, 2016, Proceedings 27 (pp. 3-17). Springer International Publishing.

---

### Author Rebuttal · Authors · 2024-11-21

**Response to Reviewer oJ2Z:**

**Response to Q1:**

To respond to the reviewer’s first question about the proof of theorem 2, we provide an analogous proof for Rademacher complexity in the appendix on page 16. We restate theorem 2 as theorem 12 and proceed to provide the derivation that follows from prior results on contraction inequalities for Rademacher/Gaussian complexity of vector-valued functions [4]. We can obtain the result in theorem 2 by bounding Rademacher complexity in theorem 12 by the Gaussian complexity. We will make sure to clarify this in the paper.

[4] Maurer, A. (2016). A vector-contraction inequality for rademacher complexities. In Algorithmic Learning Theory: 27th International Conference, ALT 2016, Bari, Italy, October 19-21, 2016, Proceedings 27 (pp. 3-17). Springer International Publishing.

---

### Meta-Review · Area_Chair_xtiR · 2024-12-14

**Recommendation:** Accept
**Confidence:** 4

**Metareview:**

Most reviewers agree that the paper is interesting and provides a straightforward improvement of Gaussian complexity bounds for multilayer neural networks with weight matrices of low rank. In particular, it offers a comparison to Frobenius norm bounds and spectrally normalized cases and shows improvements (up to constant). The analysis is simple and clean, leveraging Maurer's "chain rule" for composite classes.

However, one of the reviewers has raised a concern regarding a constant (exponential in depth) that might inflate the bound which is a limitation of the work. A similar issue appears in the classical analysis of Rademacher complexity for multilayer classes. As the authors suggest, this might be an artifact of the analysis, as a similar dependence was eliminated in [2] (in the so-called "size-independent" bounds). Whether this improvement can be achieved easily or would require a completely different proof remains unclear. Nonetheless, the current submission seems to be a reasonable first step toward Rademacher-type generalization bounds with a low-rank weight matrices.

Another minor issue, as noted by a different reviewer, is that the bounds on the complexity are technically incorrect, as they involve data-dependent quantities. Instead, they should involve fixed bounds on norms. This issue is commonly resolved by applying a union bound to the final high-probability result, and I strongly suggest the authors to address this to avoid confusion.

**Paper Award:**

No